# Cofactor and glycosylation preferences for *in vitro* prion conversion are predominantly determined by strain conformation

**Cassandra M. Burke**[1], **Daniel J. Walsh**[1], **Kenneth M. K. Mark**[1], **Nathan R. Deleault**[1], **Koren A. Nishina**[1], **Umberto Agrimi**[2], **Michele A. Di Bari**[2], **Surachai Supattapone**[1,3]*

**1** Departments of Biochemistry and Cell Biology, Geisel School of Medicine at Dartmouth, Hanover, New Hampshire, United States of America, **2** Department of Veterinary Public Health and Food Safety, Istituto Superiore di Sanità, Rome, Italy, **3** Department of Medicine, Geisel School of Medicine at Dartmouth, Hanover, New Hampshire, United States of America

* supattapone@dartmouth.edu

**Data Availability Statement:** All relevant data are within the manuscript and its Supporting Information files.

## Abstract

Prion diseases are caused by the misfolding of a host-encoded glycoprotein, PrP$^C$, into a pathogenic conformer, PrP$^{Sc}$. Infectious prions can exist as different strains, composed of unique conformations of PrP$^{Sc}$ that generate strain-specific biological traits, including distinctive patterns of PrP$^{Sc}$ accumulation throughout the brain. Prion strains from different animal species display different cofactor and PrP$^C$ glycoform preferences to propagate efficiently *in vitro*, but it is unknown whether these molecular preferences are specified by the amino acid sequence of PrP$^C$ substrate or by the conformation of PrP$^{Sc}$ seed. To distinguish between these two possibilities, we used bank vole PrP$^C$ to propagate both hamster or mouse prions (which have distinct cofactor and glycosylation preferences) with a single, common substrate. We performed reconstituted sPMCA reactions using either (1) phospholipid or RNA cofactor molecules, or (2) di- or un-glycosylated bank vole PrP$^C$ substrate. We found that prion strains from either species are capable of propagating efficiently using bank vole PrP$^C$ substrates when reactions contained the same PrP$^C$ glycoform or cofactor molecule preferred by the PrP$^{Sc}$ seed in its host species. Thus, we conclude that it is the conformation of the input PrP$^{Sc}$ seed, not the amino acid sequence of the PrP$^C$ substrate, that primarily determines species-specific cofactor and glycosylation preferences. These results support the hypothesis that strain-specific patterns of prion neurotropism are generated by selection of differentially distributed cofactors molecules and/or PrP$^C$ glycoforms during prion replication.

## Author summary

According to the "protein-only hypothesis," mammalian prions are unconventional infectious agents that lack replicating nucleic acids and instead contain misfolded forms of a host glycoprotein termed PrP$^{Sc}$. Paradoxically, despite lacking independent genomes, prions can exist as distinct self-propagating "strains," each of which is associated with its own

**Funding:** This work was supported by NIH grants R01NS102301, R56NS094576, and R21NS099928 to SS and T32AI007519 to CB. The funders had no role in study design, data collection and analysis, decision to publish, or preparation of the manuscript.

**Competing interests:** The authors have declared that no competing interests exist.

PrP$^{Sc}$ conformation and biological properties, including unique patterns of brain targeting (neurotropism) and PrP$^{Sc}$ glycosylation. The mechanism by which different PrP$^{Sc}$ conformers can cause distinct patterns of neurotropism and PrP$^{Sc}$ glycosylation is unknown, and represents an important challenge for the protein-only hypothesis. Here, we show that the prion strain conformation plays a dominant role in determining which cofactor molecules and glycosylated substrate molecules can be used to form PrP$^{Sc}$ in chemically defined biochemical assays. These results provide the first direct evidence that the major strain properties of infectious prions, including neurotropism, can be explained by the process of selective cofactor and substrate usage during PrP$^{Sc}$ replication. This concept may also explain the specific patterns of neurotropism observed for several other prion-like neurodegenerative diseases, such as Parkinson's disease and Alzheimer's disease.

## Introduction

Prion diseases are fatal neurodegenerative diseases caused by unique infectious agents termed prions. Prion diseases are invariably fatal and affect humans and other mammals. Unlike conventional infectious agents, prions lack a nucleic acid genome, and instead are formed by the autocatalytic conversion of PrP$^{C}$, a conserved host-encoded glycoprotein, into a collection of misfolded conformers, collectively termed PrP$^{Sc}$[1].

Prions can exist as strains that represent distinct conformations of PrP$^{Sc}$[2, 3]. Prion strains are characterized by specific PrP$^{Sc}$ glycosylation profiles (i.e. the relative distribution of di-, mono-, and unglycosylated PrP$^{Sc}$ molecules)[4, 5], as well as distinct patterns of PrP$^{Sc}$ accumulation in different regions of the brain leading to specific clinical phenotypes[6, 7]. How unique PrP$^{Sc}$ conformations give rise to specific strain properties remains a mystery. Interestingly, the same PrP$^{Sc}$ amino acid sequence is capable of producing several different prion strains, all representing different PrP$^{Sc}$ conformations within the same host species[8, 9], indicating that prion strain properties must be encoded by factors other than PrP sequence.

While the exact mechanism of prion replication remains unknown, cofactor molecules and PrP$^{C}$ glycosylation are both thought to play important roles in maintaining the biological infectivity[10–15] and strain properties of mammalian prions[16–21]. Using the serial protein misfolding cyclic amplification (sPMCA) technique[22, 23], Nishina *et al.* discovered that PrP$^{C}$ glycosylation preferences for prion propagation *in vitro* appear to be species-dependent [24]. Specifically, propagation of five different strains of mouse (Mo) prions requires unglycosylated (UN) mouse PrP$^{C}$ substrate, while diglycosylated (DI) mouse PrP$^{C}$ is unable to propagate mouse prions[24]. Remarkably, hamster (Ha) prions appear to have the exact opposite preferences: DI hamster PrP$^{C}$ substrate is required to propagate three different strains of hamster prions, while UN hamster PrP$^{C}$ actually inhibits propagation[24]. Hamster and mouse prions also appear to have different cofactor preferences for propagation *in vitro*[25]. Deleault *et al.* showed that the lipid cofactor phosphatidylethanolamine (PE) can be used to propagate mouse prions[11], while hamster prions can also be propagated using polyanionic molecules, such as single-stranded RNA[10]. Taken together, these studies underscore the importance of the prion replication environment to the propagation of unique prion strains.

However, the molecular basis for these species-specific preferences for prion conversion are unknown. In principle, specificity could be primarily determined by the amino acid sequence of the PrP$^{C}$ substrate or by the conformation of the PrP$^{Sc}$ seed. Distinguishing between these possibilities requires testing the ability of different PrP$^{Sc}$ strains to convert a single common

substrate, for example, by using mouse and hamster PrP<sup>Sc</sup> seeds to convert hamster PrP<sup>C</sup>. However, these experiments are not possible because transmission barriers exist between many different species[26]. These barriers to prion conversion are thought to reflect differences in amino acid sequence complementarity between the PrP<sup>C</sup> substrate and PrP<sup>Sc</sup> seed, as well as the conformation of the PrP<sup>Sc</sup> seed[27]. However, transmission barriers do not prevent cross-propagation of all prion strains, and certain organisms are more receptive to infection by different prion strains than others.

The European bank vole has emerged as a highly receptive animal model of prion diseases that can accommodate prion strains with a wide variety of PrP amino acid sequences[28–32]. Interestingly, the strain properties of prions from other species do not change when they are transmitted in transgenic mice expressing bank vole PrP[31]. Here, we use the bank vole model as a single common substrate in sPMCA to determine whether species-specific cofactor and glycosylation preferences are primarily determined by the PrP<sup>C</sup> substrate or PrP<sup>Sc</sup> seed in the absence of a transmission barrier, informing how unique conformations of PrP<sup>Sc</sup> give rise to strain properties.

## Results

### Bank vole brain homogenate is a versatile substrate

To identify a single substrate that could be used to propagate both mouse (Mo) and hamster (Ha) prion strains *in vitro*, we tested the susceptibility of crude BV (bank vole) BH (brain homogenate) substrate to mouse RML and hamster 139H seeds in sPMCA (serial Protein Misfolding Cyclic Amplification). The results show that self-propagating PrP<sup>Sc</sup> products were successfully produced in reactions containing bank vole brain homogenate seeded with either RML or 139H, but not in unseeded control reactions (Fig 1, bottom row). In contrast, mouse brain homogenate could propagate only RML (Fig 1, top row), and hamster brain homogenate could propagate only 139H (Fig 1, middle row). Taken together, these results indicate that bank vole PrP<sup>C</sup> is a uniquely susceptible single substrate for both hamster and mouse prion strains, which could be used in subsequent experiments to directly compare the cofactor and glycosylation preferences of prion strains from different species under identical reaction conditions.

To confirm that 139H and RML maintain different strain properties in bank voles, we performed three serial passages of each strain in wild type voles (M109) by intracerebral inoculation. Survival times from the third pasages were 69±4 (n = 10) for RML and 112±2 (n = 9) for 139H. In 139H-infected voles, spongiform neurodegeneration was prominent in all areas analyzed, except for the cerebellum. Voles inoculated with RML had a distinct lesion profile, characterized by a lower spongiform change in the medulla oblongata, superior colliculi and hypothalamus, while cortices were more severely involved (S1 Fig). These *in vivo* data confirm that 139H and RML display and maintain different strain properties in bank voles, including distinct patterns of neurotropism.

### Cofactor preference is determined by prion seed rather than PrP<sup>C</sup> substrate

To distinguish whether cofactor preference for PrP<sup>Sc</sup> formation is primarily determined by the PrP<sup>C</sup> substrate or the input prion seed, we first employed RNase to specifically degrade RNA cofactor molecules in crude brain homogenate substrates. To ensure the efficacy of the RNase treatment, RNA levels were quantified in treated and untreated brain homogenate substrates (S1 Table). Removal of single-stranded RNA molecules by pretreatment of crude brain homogenate with RNase had no effect on sPMCA reactions containing either mouse or bank vole substrate seeded with mouse prion strains RML or Me7 (Fig 2, rows 1–2 and 5–6), but

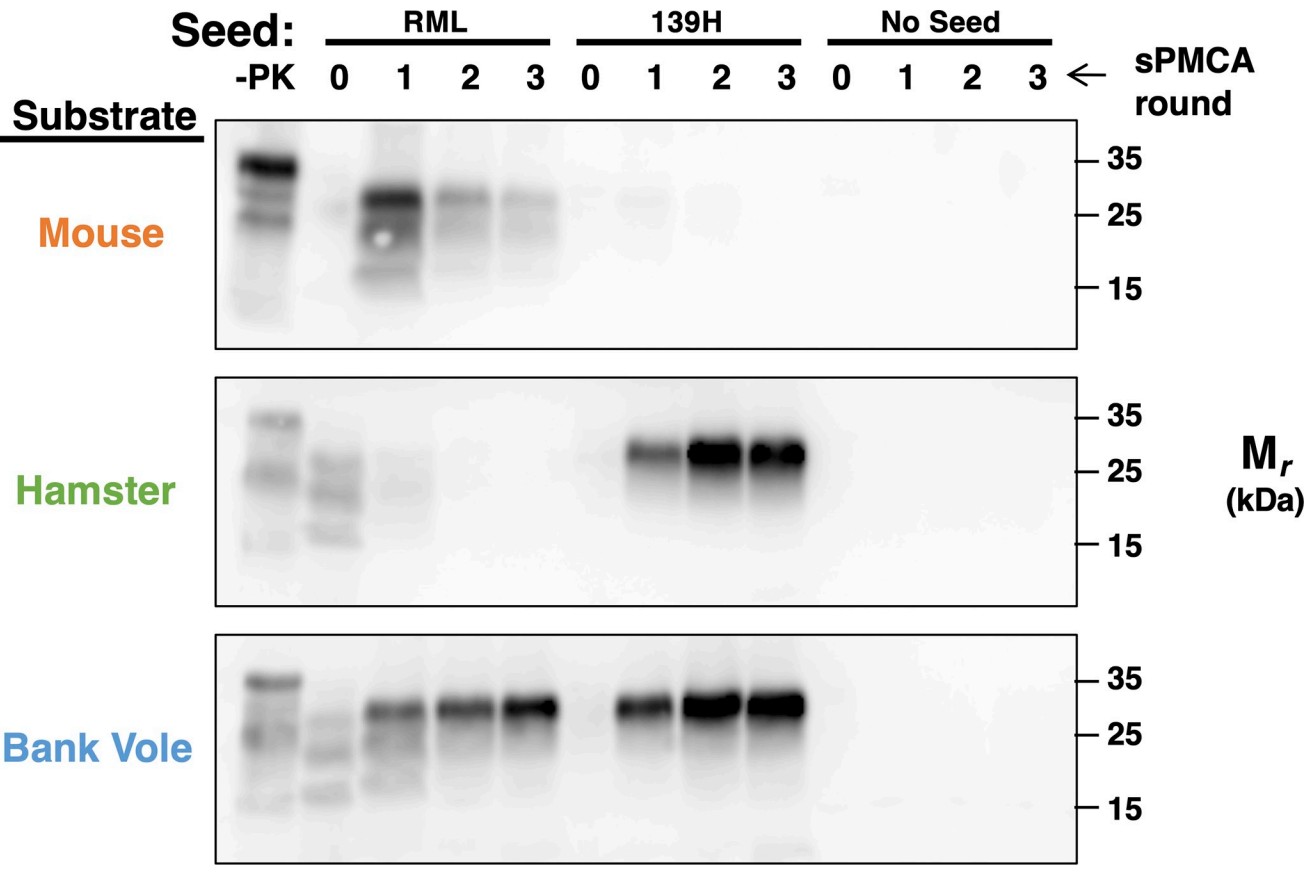

**Fig 1. Seed-dependent conversion of bank vole, hamster, and mouse brain homogenate substrates.** Western blots showing three-round sPMCA reactions using normal brain homogenates from different species as the substrates and seeded with mouse (RML) or hamster (139H) prion-infected brain homogenates, as indicated. Day 0 samples are a seeded reaction not subject to sonication. -PK = samples not subjected to proteinase K digestion; all other samples were proteolyzed.

inhibited reactions containing either hamster or bank vole substrate seeded with hamster prion strains 139H and Sc237 (Fig 2, rows 3–4 and 7–8). These results suggest that RNA molecules are disposable for propagation of the mouse prion strain regardless of PrP$^C$ substrate sequence, while RNA molecules are the preferred cofactor for propagation of hamster prion strains, regardless of PrP$^C$ substrate sequence.

As a complementary approach to determine the cofactor preferences for PrP$^{Sc}$ formation *in vitro*, we also performed sPMCA reactions using immunopurified PrP$^C$ (S2 Fig) supplemented with purified cofactor molecules. The results show that sPMCA reactions containing either mouse or bank vole PrP$^C$ substrate seeded with RML prions could successfully propagate with purified phospholipid but not synthetic poly(A) RNA cofactor (Fig 3, rows one and three). In contrast, reactions containing ether hamster or bank vole PrP$^C$ substrate seeded with 139H prions could propagate with either cofactor (Fig 3, rows two and four). These results support the conclusion that cofactor preference for PrP$^{Sc}$ formation *in vitro* is selected by the conformation of the prion seed rather than the sequence of the PrP$^C$ substrate.

We conducted similar studies with the additional mouse prion strains, 139A, 22A, and 301C. Like RML, none of these strains could use poly(A) RNA as a cofactor in sPMCA reactions containing mouse PrP$^C$ substrate (Fig 4, column 2). Interestingly, PrP$^{Sc}$ molecules were produced by the third round in reactions containing bank vole PrP$^C$ and poly(A) RNA for all

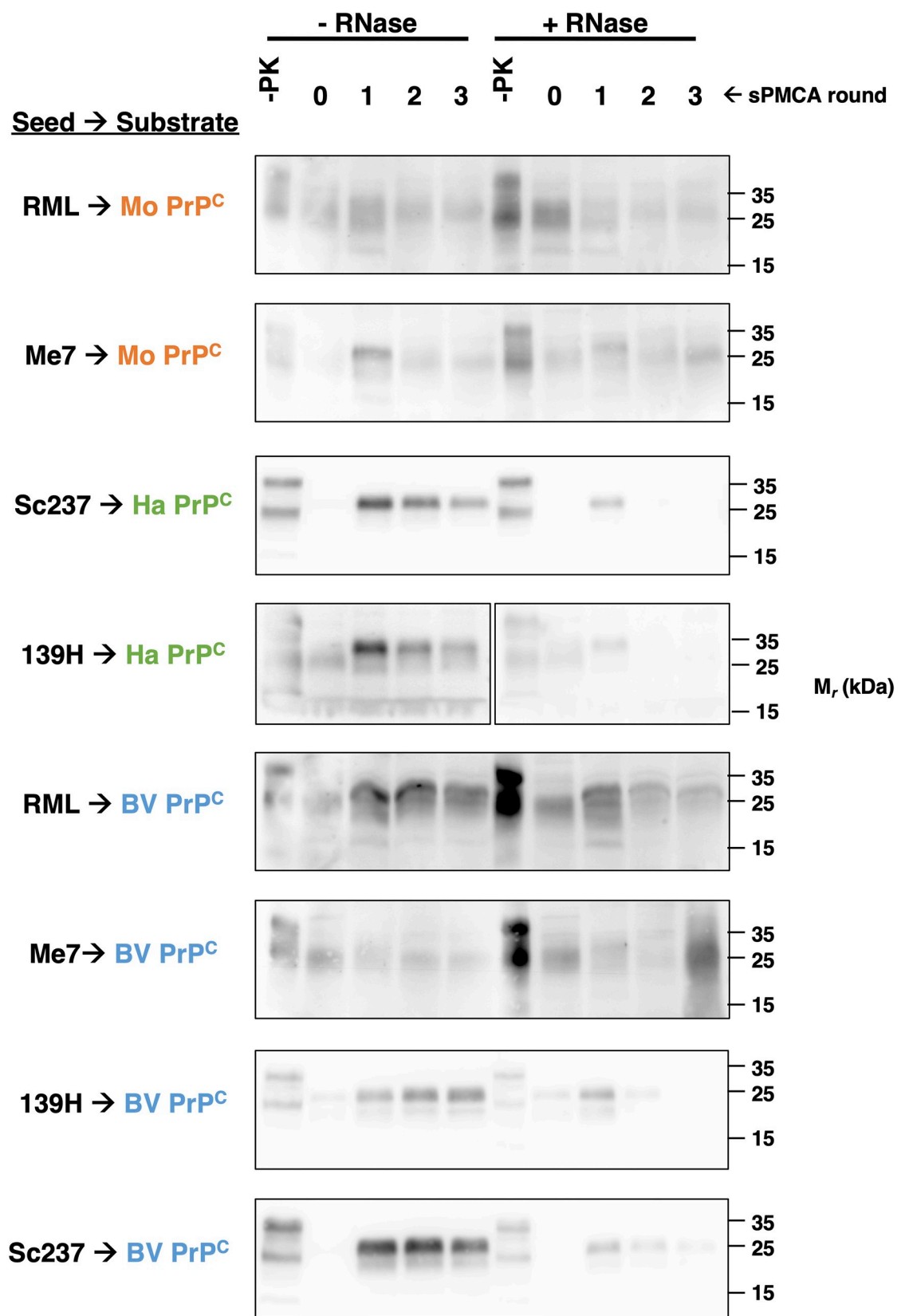

**Fig 2. Effect of RNase treatment on PrP$^{Sc}$ propagation *in vitro*.** Western blots showing three-round sPMCA reactions using either BV, hamster or mouse crude brain homogenate substrates seeded with the indicated prion-infected brain homogenate, or no seed. Where indicated, the crude brain homogenate was pretreated with RNase prior to sPMCA (+RNase). Day 0 samples are a seeded reaction not subject to sonication. -PK = samples not subjected to proteinase K digestion; all other samples were proteolyzed. Note that the PMCA products in this experiment appear generally weaker compared to other experiments (e.g. Fig 1). This is most likely because the crude brain homogenate substrates (including mock-treated samples) were subjected to pre-incubation at 37˚C prior to addition of seed, allowing endogenous enzymes to partially degrade and/or misfold PrP$^{C}$ substrate. Nonetheless, all of the mock-control samples appear to successfully maintain the lower levels of PrP$^{Sc}$ over three successive PMCA rounds.

three strains. However, the mobility of the PK-resistant core of the PrP$^{Sc}$ molecules (~27 kDa) formed in all three RNA-supplemented reactions was significantly faster than those of the PrP$^{Sc}$ molecules formed in reactions supplemented with crude PrP$^{0/0}$ brain homogenate (~30 kDa) (Fig 4, columns 3 and 4, compare black arrowheads versus white arrowheads). These results suggest that, regardless of PrP$^{C}$ substrate sequence, the mouse prion strains 139A, Me7, 22A and 301C do not prefer to use RNA as a cofactor. However, bank vole PrP$^{C}$ is capable of using RNA to form an alternative PrP$^{Sc}$ conformer in a relatively inefficient adaptation process. This adaptation process, which is caused by propagation with a purified cofactor and we term cofactor restriction can also be induced by phospholipid cofactor using recombinant (rec)PrP[16], and has been observed before with bank vole PrP substrates [18].

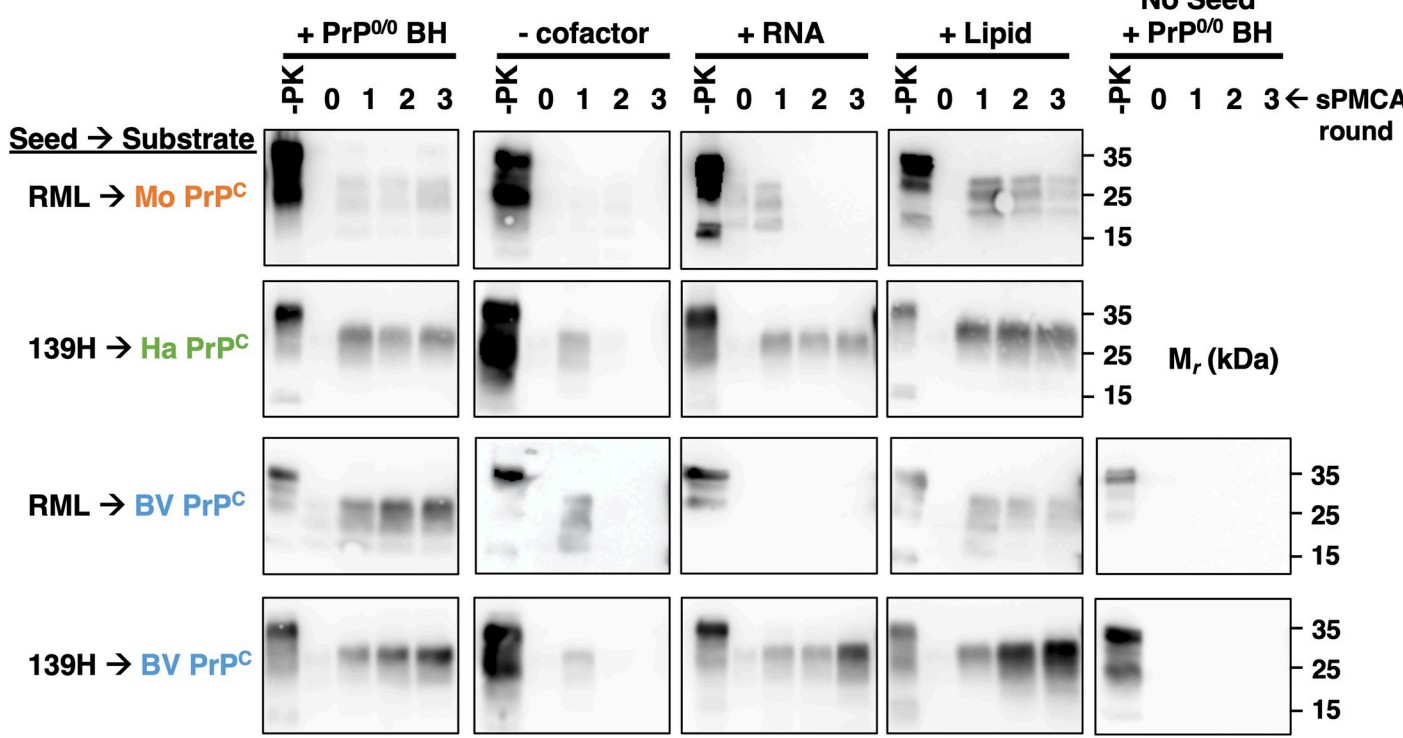

**Fig 3. Seed-dependent cofactor utilization of bank vole, hamster, and mouse PrP$^{C}$ substrates.** Western blots showing three-round sPMCA reactions using immunopurified PrP$^{C}$ substrates and seeded with the indicated prion-infected brain homogenate or no seed. Reactions were supplemented with either PrP$^{0/0}$ brain homogenate, PBS containing 1% Triton X-100 (- cofactor), poly(A) RNA (RNA), or a lipid cofactor preparation (lipid). Day 0 samples are a seeded reaction not subject to sonication. -PK = samples not subjected to proteinase K digestion; all other samples were proteolyzed. Note: in some blots, PrP$^{Sc}$ amplification can be seen on Day 1 either in the absence of cofactor or with an inappropriate cofactor. This may be due to the presence of cellular cofactors in the diluted prion-infected brain homogenates used as Day 1 seeds.

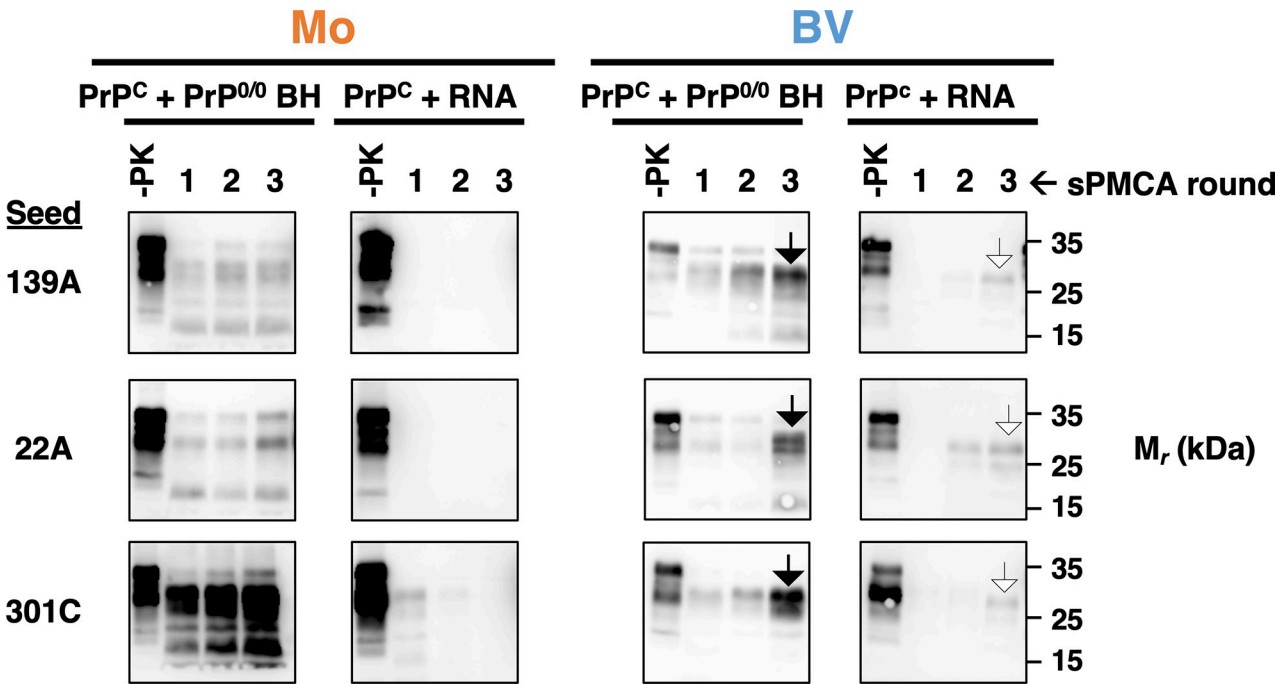

**Fig 4. Conversion of mouse prion strains using RNA as a cofactor.** Western blots of three-round sPMCA reactions using either bank vole or mouse immunopurified PrP$^C$ substrates supplemented with either PrP$^{0/0}$ brain homogenate or poly(A) RNA (RNA) and seeded with the indicated prion-infected mouse brain homogenate or no seed. Black arrowheads highlight a migration pattern at ~30 kDa and white arrowheads highlight a migration pattern at ~27 kDa. Day 0 samples are a seeded reaction not subject to sonication. -PK = samples not subjected to proteinase K digestion; all other samples were proteolyzed.

Finally, we also investigated the cofactor preferences for a pair of matched recPrP$^{Sc}$ conformers which are derived from the same original seed, but subsequently propagated either in the presence or absence of phospholipid cofactor, and therefore termed "cofactor" and "protein-only" PrP$^{Sc}$, respectively[16, 33, 34]. The results show that both conformers are able to use either purified phospholipid or poly(A) RNA as cofactors in sPMCA reactions using immuno-purified native bank vole PrP$^C$ substrate (Fig 5, columns three and four). However, cofactor PrP$^{Sc}$-seeded reactions using poly(A) RNA were relatively inefficient (Fig 5, top row, third column), and produced a PrP$^{Sc}$ product that differed in pattern and mobility from the products of reactions containing either crude PrP$^{0/0}$ brain homogenate or purified phospholipid cofactor (Fig 5, top row, compare 3rd column black arrowhead to 1$^{st}$ column white arrowhead and 4th column grey arrowhead). In contrast, all protein-only PrP$^{Sc}$-seeded reactions efficiently produced similar sPMCA products regardless of the type of cofactor used during propagation (Fig 5, second row, 1$^{st}$, 3$^{rd}$, and 4$^{th}$ columns). These results indicate that the cofactor and protein-only PrP$^{Sc}$ conformers have different cofactor preferences. In particular, RNA does not appear to be an efficient cofactor for cofactor PrP$^{Sc}$, and use of RNA as a cofactor seems to induce conformational adaptation of cofactor PrP$^{Sc}$ but not protein-only PrP$^{Sc}$.

## Glycosylation preference is also determined by prion seed rather than PrP$^C$ substrate

Since we observed that prion seeds determine species-specific cofactor preferences for PrP$^{Sc}$ formation, we hypothesized that prion seeds may also determine species-specific glycosylation preferences for PrP$^{Sc}$ formation. To test this hypothesis, we performed sPMCA reactions

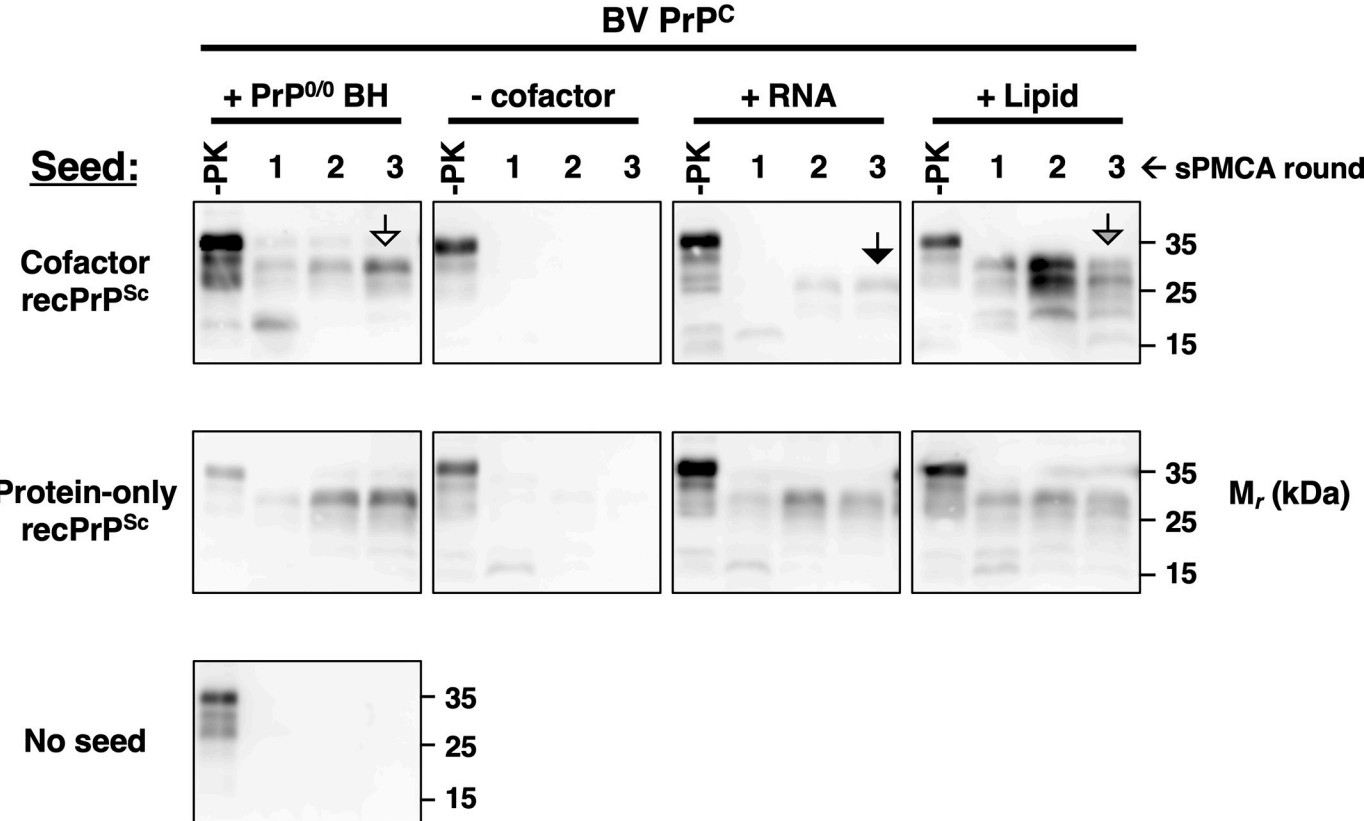

**Fig 5. Cofactor utilization of recombinant PrP$^{Sc}$ conformers.** Western blots showing three-round sPMCA reactions using immunopurified bank vole PrP$^C$ as the substrate and seeded with either cofactor recPrP$^{Sc}$, protein-only recPrP$^{Sc,}$ or no seed. Reactions were supplemented with either PrP$^{0/0}$ brain homogenate, PBS containing 1% Triton X-100 (- cofactor), poly(A) RNA (RNA), or a lipid cofactor preparation (lipid). Black, white, and grey arrowheads highlight distinct PrP$^{Sc}$ migration patterns. Day 0 samples are a seeded reaction not subject to sonication. -PK = samples not subjected to proteinase K digestion; all other samples were proteolyzed.

partially purified native PrP$^C$ substrates containing diglycosylated PrP$^C$ (DI) or unglycosylated PrP$^C$ (UN) supplemented with PrP$^{0/0}$ brain homogenate (S3 Fig). The results show that reactions seeded by RML prions propagate successfully with UN but not DI substrate (for both bank vole and mouse) (Fig 6, first and third rows). In contrast, reactions seeded by 139H prions propagate successfully with DI substrate (for both bank vole and hamster) (Fig 6, second and fourth rows). Interestingly, bank vole UN, but not hamster UN, can also be converted by 139H seed (Fig 6, second and fourth rows, right-hand images). However, on round three of the sPMCA reactions containing bank vole UN, the mobility of the PK-resistant core of the PrP$^{Sc}$ molecules formed in the reactions was significantly faster than the mobility of the PK-resistant cores formed during rounds 1 and 2 (Fig 6, fourth row, right-hand image). This mobility shift was reproducibly observed in two additional independent experiments (S4 Fig). Taken together, these results show that, as with cofactor molecules, glycosylation preference is mainly determined by the input prion strain, not the PrP$^C$ substrate sequence, and that strains may undergo conformational adaptation to propagate using non-preferred glycoform substrates.

## Discussion

In this study, we leveraged the broad susceptibility of bank vole PrP$^C$ to perform reconstituted sPMCA experiments with PrP$^{Sc}$ seeds from different animal species in a single, common

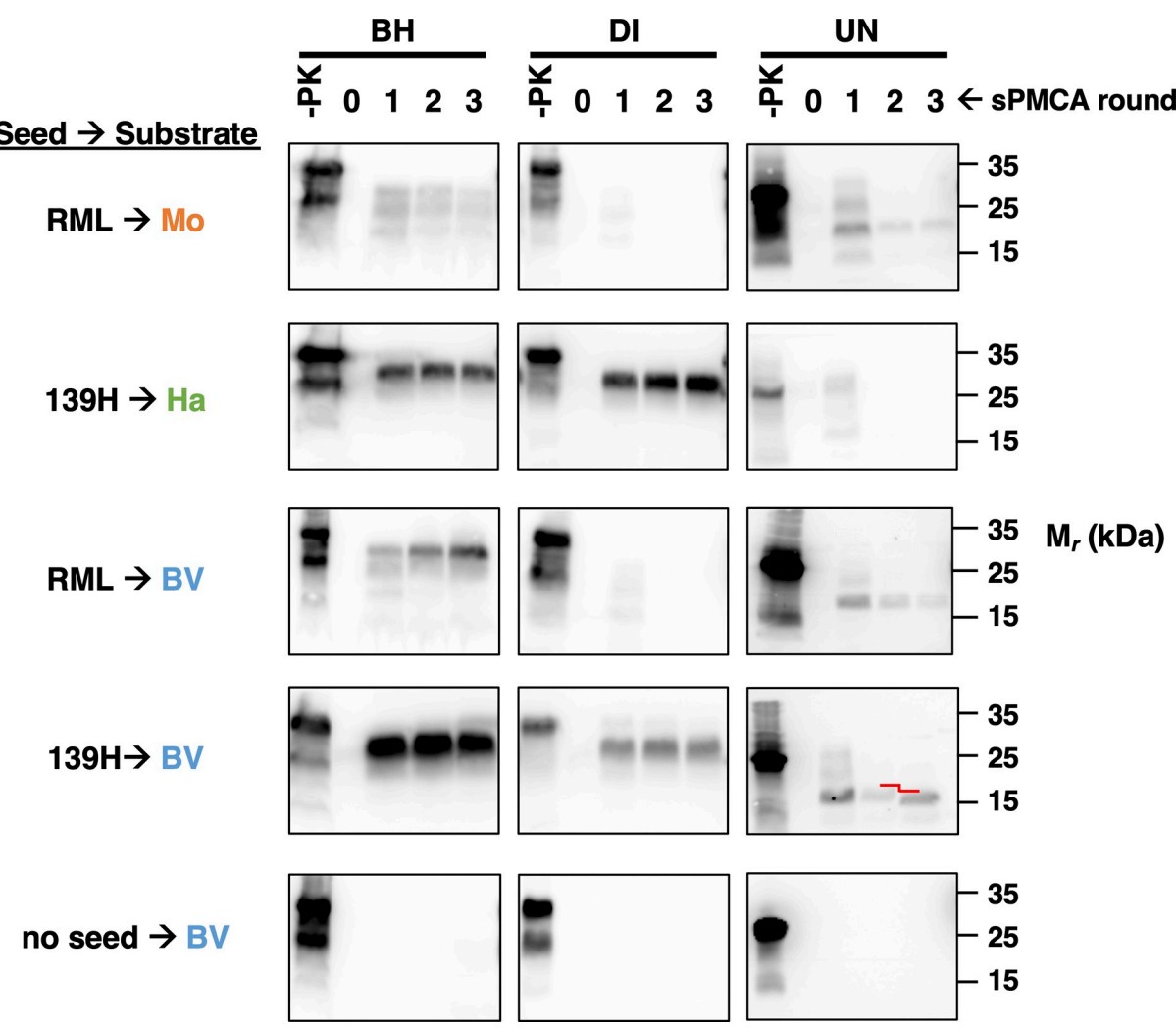

**Fig 6. Seed-dependent conversion of bank vole, hamster, and mouse PrP$^C$ substrates containing specific glycoforms.** Western blots showing three-round sPMCA reactions using normal brain homogenate (BH), partially-purified diglycosylated PrP$^C$ (DI) supplemented with PrP$^{0/0}$ brain homogenate, or partially purified unglycosylated PrP$^C$ (UN) from various species supplemented with PrP$^{0/0}$ brain homogenate as the substrate. Reactions were seeded with various prion-infected brain homogenates, as indicated. The red lines highlight a shift in the apparent MW of the day three sample of 139H-seeded bank vole UN PrP$^C$. Day 0 samples are a seeded reaction not subject to sonication. -PK = samples not subjected to proteinase K digestion; all other samples were proteolyzed.

substrate. Our most striking finding is that both cofactor and PrP$^C$ glycoform preferences appear to be selected primarily by PrP$^{Sc}$ seed conformation rather than PrP$^C$ sequence. This is most clearly illustrated by the behavior of RML in propagation reactions containing either mouse or bank vole PrP$^C$ substrates. In either species, RML can propagate efficiently with phospholipid cofactor and unglycosylated PrP$^C$ substrate, but not with RNA cofactor or diglycosylated PrP$^C$ substrate. Another good example is the ability of RNase treatment to inhibit propagation of Sc237 and 139H (both not RML or Me7) in both hamster and bank vole brain homogenates. Thus, the prion strain selects its preferred cofactor and PrP$^C$ glycoform without regard for PrP$^C$ sequence. We hypothesize that both cofactor molecules and glycans shape the potential folding landscape of PrP molecules, thereby allowing or even favoring the formation of specific PrP$^{Sc}$ conformations.

In some cases, we observed that prion strains appeared to use non-preferred cofactors or PrP$^C$ glycoforms to convert bank vole PrP$^C$ into PrP$^{Sc}$: (1) several mouse prion strains induced bank vole PrP conversion when the non-preferred cofactor, RNA was used (Fig 4), (2) 139H induced conversion of bank vole UN, the non-preferred PrP$^C$ glycosylation substrate (Fig 6), and (3) cofactor recPrP$^{Sc}$, a synthetic strain produced with lipid cofactor PE[11], could employ RNA cofactor molecules to convert bank vole PrP (Fig 5). However, in each of these cases, propagation was inefficient (i.e. sPMCA product detectable only after 2–3 rounds) and the PK-resistant core of the PMCA products were different in size from those of the original PrP$^{Sc}$ seeds. These observations indicate that prion strains undergo adaptation into new PrP$^{Sc}$ conformers when they are forced to propagate with sub-optimal cofactor molecules or PrP$^C$ glycoforms[4, 35, 36]. This interpretation is supported by the observation that *in vitro* propagation of the hamster prion strain 263K under RNA-depleted conditions produced infectious prions with novel strain properties *in vivo*[17]. Our interpretation is also consistent with prior work showing that cofactor molecules [11, 17, 18, 20] and PrP$^C$ glycosylation [12, 21, 37–40] can influence prion strain properties. More broadly, the observation that sub-optimal cofactor and PrP$^C$ glycoform usage leads to PrP$^{Sc}$ adaptation lends support to our main conclusion that each prion strain and its accompanying PrP$^{Sc}$ conformer has specific cofactor and PrP$^C$ glycosylation preferences.

Cofactor preferences are relative, and the ability of a single prion strain to use different cofactors appears to be concentration-dependent. This is most clearly illustrated by the successful propagation of Sc237 and 139H in purified reactions with high concentrations of purified brain phospholipids, despite the inability of these same strains to propagate efficiently in RNase-treated brain homogenates, which contain phospholipids in lower concentrations. In contrast, both RML and Me7 can propagate efficiently in RNase-treated brain homogenates, indicating that brain phospholipids are more potent cofactors for these strains than for Sc237 and 139H. Brain phospholipids have also been found to inhibit chronic wasting disease (CWD) PrP$^{Sc}$-seeded amyloid formation [41] (and conversely, RNA molecules can inhibit the propagation of or cause adaptation of a mouse recombinant PrP$^{Sc}$ conformer that uses phospholipid cofactor (S5 Fig)) suggesting that molecules which serve as cofactors for one prion strain may actually inhibit conversion of another strain.

Although PrP$^{Sc}$ strain comformation appears to play the dominant role in cofactor and glycoform selection for the versatile bank vole PrP$^C$ substrate, it seems likely that the primary sequence of PrP$^C$ could play a more important role in limiting cofactor and glycoform usage other species. This could explain why all 5 of mouse strains that we studied appear to be resistant to RNase treatment, whrease both of the hamster strains tested were RNase-sensitive. Interestingly, bank vole PrP$^C$ is natural chimera of the mouse and hamster PrP$^C$ sequences, and it is possible that some of the residues that differ between the mouse and hamster sequences are responsible for facilitating specific cofactor and glycoform preferences.

There are several limitations of our study. First, we were restricted to using previously identified cofactor molecules (phospholipid and RNA)[11, 25]. There may be other, undiscovered endogenous cofactors that facilitate the replication of specific prion strains, and some strains may even utilize multiple cofactors[20]. Second, our study was limited to a small number of rodent species, so it is possible that these findings are not generalizable to prions from other animal species. Nonetheless, it is likely that the fundamental mechanism of infectious prion replication is likely to be conserved among mammalian species. Third, we only used biochemical assays for our experiments, and therefore cannot say conclusively whether species-specific cofactor and glycosylation preferences are determined by the same mechanism *in vivo*. However, the ability of the prion seed to select its preferred glycoform substrate *in vitro* provides a

logical explanation for the observation that prion strains can maintain a characteristic PrP<sup>Sc</sup> glycoform ratio *in vivo*, even following transmission into different animal species[31, 42].

Our *in vitro* results also offer a logical explanation for the *in vivo* phenomenon of neurotropism, i.e. the strain-specific patterns of regional PrP<sup>Sc</sup> deposition and vacuolation seen in prion-infected brains. Specifically, our results support the cofactor selection model of neurotropism (Fig 7), originally proposed by Geoghegan *et al.*[16, 43][44]]. The results of the *in vitro* experiments in this manuscript are summarized on the left side of Fig 7, where the conformations of different PrP<sup>Sc</sup> seeds are maintained through use of their preferred cofactor molecule for replication. The right side of Fig 7 models how this concept translates *in vivo*. We envision that specific cofactor molecules are differentially distributed throughout the brain, causing specific prion strains to preferentially replicate in brain regions containing higher levels of their preferred cofactor molecule. This model builds upon abundant evidence that cofactor molecules play important roles in prion replication and biological infectivity[11, 25, 34, 43]. To test the cofactor selection model, it will be necessary to first identify cofactor molecule(s) that are able to maintain the strain properties of specific prion strains. In prior work, we showed that PE alone cannot maintain the strain properties of RML, Me7, or 301C[16].

Other hypotheses have been proposed to explain the phenomenon of neurotropism, but they each have drawbacks. The selective degradation hypothesis proposes that different prion strains are differentially and specifically degraded in distinct regions of the brain. While the rate of prion formation and clearance has been shown to influence strain tropism at an organ level (i.e. tropism to the brain and spleen)[45], no mechanisms for strain-specific degradation have been identified to date. The PrP<sup>Sc</sup> glycosylation hypothesis proposes that neurons distinguish between different prion strains based on their PrP<sup>Sc</sup> glycosylation status. This hypothesis is attractive for several reasons: (1) prion strains exhibit specific, relative distributions of di-, mono-, and unglycosylated PrP<sup>Sc</sup> molecules[5], (2) glycans are large and located on the surface of PrP, ideally positioned as intermolecular targets[46], and (3) there are a large number of possible glycosylation variants, meaning there could, in theory, be many different opportunities for specific interactions[47]. However, it has been shown that PrP<sup>Sc</sup> glycosylation is not absolutely required for strain-dependent prion neurotropism, as unglycosylated RML and 301C inoculum maintain their strain-specific PrP<sup>Sc</sup> distribution and vacuolation[48]. Additionally, we recently showed that a recombinant PrP<sup>Sc</sup> conformer lacking glycosylation displays same pattern of neurotropism as the same strain of native PrP<sup>Sc</sup>[34]. Finally, the "coprion" hypothesis[49] posits that small, replicating nucleic acids encode prion strain information, but no such molecules have ever been discovered. In contrast, the cofactor selection model proposes that cofactors are drawn from pre-existing cellular pools and are not replicating. This model broadens the possibilities for the types of molecules that can serve as cofactors as well as the mechanisms by which such molecules can influence strain properties.

Our data are consistent with previous reports that PrP<sup>C</sup> glycosylation status can modify *in vitro* prion conversion[50–52], and also support a model in which selection of PrP<sup>C</sup> glycoforms could also influence neurotropism[19, 38, 53]. This model is analogous to the cofactor selection model, but is based on natural variation in the regional distribution of PrP<sup>C</sup> glycoforms rather than cofactor molecules. Consistent with this model, it has been shown that specific PrP<sup>C</sup> glycoforms are enriched in different brain regions[53, 54], specific PrP<sup>Sc</sup> strains are associated with strain-specific glycoform ratios[55], and that a synthetic prion strain (SSLOW) can use unglycosylated hamster recPrP as a substrate [20]. Furthermore, transgenic mice expressing specific PrP<sup>C</sup> glycoforms display different patterns of neuropathology compared to WT mice when inoculated with the same prion strain [38]. Factors other than cofactor molecules and PrP<sup>C</sup> glycosylation may also influence neurotropism. For instance, different prion strains have been shown to utilize different endocytic routes for infection, and it is possible

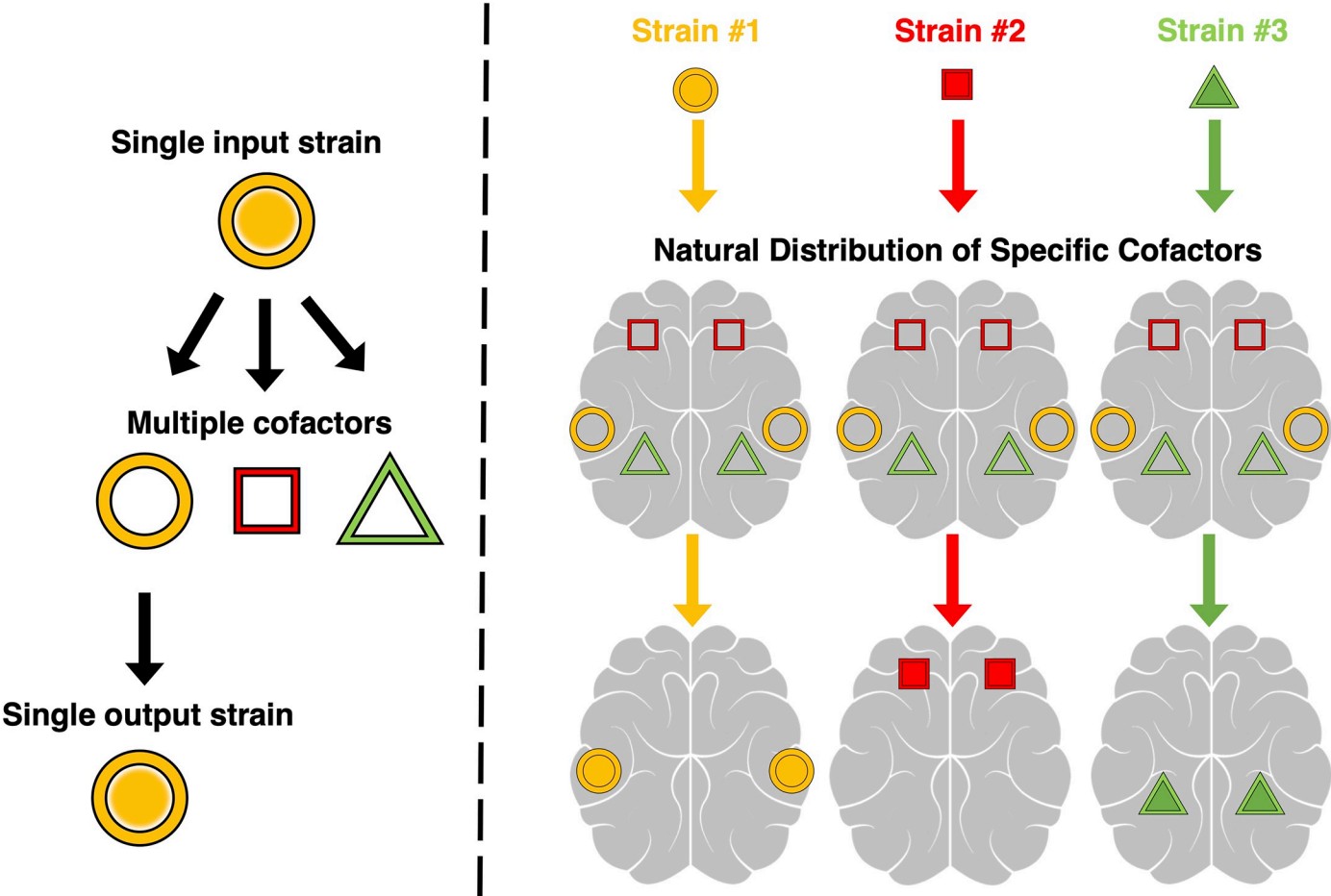

**Fig 7. Schematic of The Cofactor selection model of neurotropism.** Left: A single input prion strain (yellow circle) preferentially uses a certain cofactor molecule (yellow ring). When replicating in an environment that contains multiple other potential cofactor molecules (red square and green triangle), the input prion strain selects its preferred cofactor for propagation, leading to faithful strain replication. Right: Cofactor molecules are distributed differentially throughout the brain. Different prion strains with different cofactor preferences selectively replicate in regions enriched for their preferred cofactor molecule, leading to unique strain pathologies (bottom).

that cell-specific differences in endocytic mechanisms could contribute to neurotropism[56]. Also, Hu *et al.* showed that crude homogenates of different brain regions do not display differential susceptibility to different prion strains in PMCA reactions[57]. One possible explanation for this result is that a wide variety of potential cofactor molecules and PrP$^C$ glycoforms may be available in crude brain homogenates, but only a much more limited and specific subset may be accessible within each brain region *in vivo*. Thus, multiple cellular factors may work in tandem to generate strain specific patterns of neurotropism.

In conclusion, we report for the first time that species-specific cofactor and glycosylation preferences are primarily determined by the conformation of the PrP$^{Sc}$ seed, rather than the amino acid sequence of the PrP$^C$ substrate. Based on our findings, we suggest that the mechanism of strain-dependent neurotropism depends cofactor and PrP$^C$ glycoform selection by the template PrP$^{Sc}$ conformer during self-replication. This model explains how neurotropism and other biological strain properties can be associated with specific PrP$^{Sc}$ conformations [58–64]. It is intriguing to consider the role that cofactor molecules and post-translational modifications may play in the specific regional distribution patterns seen in other neurodegenerative

diseases associated with prion-like mechanisms. For instance, tau filaments from the brains of patients with chronic traumatic encephalopathy (CTE) appear to contain a hydrophobic molecule within their core structure[65], whereas tau filaments from Alzheimer's disease (AD) patients lack such a molecule. Interestingly, AD filaments have a different molecular structure than CTE filaments[65, 66] and accumulate in different regions of the brain than CTE filaments[67].

## Materials and methods

### Preparation of *in vivo* inocula

Vole brains affected with 139H and RML strains at second passages were homogenized at 10% (wt/vol) in sterile PBS prior to intracerebral inoculation into bank voles carrying methionine at codon 109.

### Animal inoculations

Groups of 8-week-old bank voles were inoculated intracerebrally with 20 μl of homogenate into the left cerebral hemisphere under ketamine anesthesia (ketamine 0.1 μg/gm). The animals were examined twice a week until neurological clinical signs appeared, after which they were examined daily. Diseased animals were culled at the terminal stage of the disease by exposure to a rising concentration of carbon dioxide but before neurological impairment compromised their welfare. Survival time was calculated as the interval between inoculation and culling or death. *Post-mortem*, the brain was removed and fixed in formalin.

### Neuropathology

Brains were trimmed at standard coronal levels, embedded in paraffin wax, cut at 6 μm and stained with haematoxylin and eosin. For the construction of lesion profiles, vacuolar changes were scored in nine grey-matter areas of the brain. Sections were randomly mixed and coded for blind pathological assessment. Vacuolation scores were derived from the examination of seven and eight voles, respectively for RML and 139H group of inocula.

### Purification of UN PrP$^C$ and DI PrP$^C$ substrates

Purification of UN PrP$^C$ and DI PrP$^C$ substrates were based on a protocol from Nishina *et al*. [24]. Six grams of brains from either bank voles with the M109 genotype, WT C57BL/6J mice, or WT Syrian hamsters were Potter homogenized in 40 mL of ice-cold Buffer A [20 mM MOPS pH 7.0 and 150 mM NaCl] containing cOmplete EDTA-free Protease Inhibitors (Roche, Basel, Switzerland). The homogenate was initially centrifuged at 200 x *g* for 30 sec; the post-nuclear supernatant was then removed and centrifuged at 3,200 x *g* for 20 min. The resulting pellet was resuspended and Dounce homogenized in 30 mL of Buffer A and cOmplete EDTA-free Protease Inhibitors. Four milliliters of a detergent mixture [10% sodium deoxycholate/ 10% Triton X-100] was added to the homogenate, and the mixture was incubated on ice for 30 min and centrifuged at 100,000 x *g* for 30 min. The solubilized supernatant was brought up to 10mM imidazole with 1M imidazole pH 7.0.

The solubilized supernatant was applied to a pre-equilibrated 2 mL IMAC-CuSO$_4$ column. The column was washed with 10 mL of IMAC-CuSO$_4$-W [Buffer A, 10 mM imidazole in Buffer A (pH 7.0), 1% Triton X-100]. The IMAC-CUSO$_4$ column was eluted with 4 mL IMAC-CuSO$_4$-E2 [20mM MOPS pH 7.5, 150mM NaCl, 150 mM imidazole in Buffer A (pH 7.0), and 1% Triton X-100] containing cOmplete EDTA-free Protease Inhibitors. The eluate was applied to a pre-equilibrated 1 mL agarose-bound wheat germ agglutinin column (Vector

Laboratories, Burlingame, CA, USA). The column was washed with 20 mL of WGAW [20mM MOPS pH 7.5, 150mM NaCl and 1% Triton X-100]. The sample was eluted with 4 mL of WGAE [20mM MOPS pH 7.5, 150mM NaCl, 200 mM N-acetylglucosamine, 1% Triton X-100] and cOmplete EDTA-free Protease Inhibitors. The eluate was loaded into a 3,500 MWCO Slide-A-Lyser (Thermo Fisher Scientific, Waltham, MA, USA) and dialyzed overnight into Buffer C [20 mM MOPS (pH 7.5), 150 mM NaCl, and 0.5% Triton X-100] to yield the PrP$^C$ product, DI.

To purify the deglycosylated PrP$^C$ substrate, UN, 50 μL of DI PrP$^C$ was combined with 5 μL (2,500 units) of glycerol-free PNGase F (New England Biolabs, Ipswich, MA, USA) and incubated at 37°C for 24 hr with shaking at 250 r.p.m. Next, at 4°C, the solution was mixed with 20 μL of agarose-bound wheat germ agglutinin (Vector Laboratories, Burlingame, CA, USA) that was pre-equilibrated in Buffer C. The solution was end-over-end rotated for 30 min at 4°C, spun for 10 sec at *500* x *g*, then the supernatant was collected and used as the final UN PrP$^C$ product.

## Immunopurification of PrP$^C$ from brain tissue

PrP$^C$ was immunopurified from brains of either bank voles with theM109 genotype, WT C57BL/6J mice, or WT Syrian hamsters based on a previously established protocol[68]. Using an electric Potter homogenizer, 12 g of brains were homogenized in 80 mL Buffer A with cOmplete Protease Inhibitors. The resulting homogenate was centrifuged at 3,200 x *g* for 25 min at 4°C. The supernatant was discarded, and the pellets were resuspended to a volume of 40 mL by Dounce homogenizing in Buffer A, 1% (w/v) sodium deoxycholate, 1% (v/v) Triton X-100. The homogenate was incubated on ice for 30 min to solubilize PrP$^C$, then centrifuged at 100,000 x *g* for 40 min at 4°C.

The solubilized supernatant was placed into a 50 mL conical tube with 1 mL of Protein A Agarose (Thermo Fisher Scientific) and end-over-end rotated for 30 min at 4°C as a pre-clear step. Next, the supernatant/Protein A mixture was poured through an Econo-Pac (Bio-Rad, Hercules, CA) column and the flow-thru was collected as the pre-cleared load.

The pre-cleared load was passed over a column packed with 2 mL Protein A Agarose resin (Pierce) cross-linked to 6D11 mAb that was pre-equilibrated with Buffer A, 1% (w/v) sodium deoxycholate, 1% (v/v) Triton X-100 at a flow rate of 0.75 mL/min. The column was washed with 36 mL of Wash Buffer 1 [20 mM Tris pH 8.0, 1% (v/v) Triton X-100, 500 mM NaCl, 5 mM EDTA], followed by 24 mL of Wash Buffer 2 [Buffer A, 0.5% (v/v) Triton X-100] at a flow rate of 1.0 mL/min. A 50 mL conical tube containing 900μL of Neutralization Buffer [1M Tris pH 9.0, 5% (v/v) Triton X-100, 1.4 M NaCl] was placed beneath the column. The column was manually eluted using a syringe filled with Elution Buffer [0.1 M glycine pH 2.5, 100 mM NaCl] until a volume of 15 mL was reached.

The eluate was brought to 50 mL with SP Equilibration/Wash Buffer [20 mM MES pH 6.4, 0.15 M NaCl, 0.5% (v/v) Triton X-100] and applied slowly to a 1.5 mL SP Sepharose (Sigma Aldrich, St. Louis, MO) ion exchange column that was pre-equilibrated with 10 column volumes of SP Equilibration/Wash Buffer. The column was washed with 15 mL of SP Equilibration/Wash Buffer And eluted with 5 mL of SP Elution Buffer [20 mM MOPS pH 7.5, 0.50 M NaCl, 1% (v/v) Triton X-100] containing cOmplete EDTA-free Protease Inhibitors. The eluate was dialyzed in 3,500 MWCO Slide-a-Lyzer cassettes (Fisher Scientific, Waltham, MA, USA) overnight against 4 L of Exchange Buffer [20 mM MOPS pH 7.5, 150 mM NaCl, 0.5% (v/v) Triton X-100].

## General serial protein misfolding cyclic amplification (sPMCA) methods

The general serial protein misfolding cyclic amplification (sPMCA) experimental method was adapted from Castilla *et al.*[69]. All PMCA reactions were sonicated in microplate horns at

37°C using a Misonix S-4000 power supply (Qsonica, Newtown, CT) set to power 70 for three rounds. One round of PMCA is equal to 24 hr. The first round of PMCA was seeded with a volume of PrP$^{Sc}$ equal to 10% of the total reaction volume. To propagate the reaction between PMCA rounds, 10% of the reaction volume was transferred into a new, unseeded, substrate mixture. Due to the sensitivity of sPMCA[70], measures were taken to prevent sample contamination. Sample conical tubes were sealed with Parafilm (Bemis Company, Oshkosh, WI) and the sonicator horn was cleaned with a solution of 10% SDS, 5% acetic acid between experiments to prevent cross-contamination. Sample conical tubes were spun at 500 x *g* for 5 sec to remove liquid from the conical tube lids before propagation and samples were propagated individually using aerosol resistant pipette tips. The experimenter wore two pairs of gloves and changed the outer layer of gloves when handling a new sample. With each experiment, a sentinel conical tube (a conical tube containing the entire sPMCA reaction mixture but lacking seed) was also placed in the sonicator horn to detect contamination.

## sPMCA with DI PrP$^C$ or UN PrP$^C$ substrates

The reconstituted sPMCA method using DI PrP$^C$ or UN PrP$^C$ substrates was adapted from Nishina *et al.*[24]. Briefly, 100 μL reactions containing 50 μL DI PrP$^C$ or UN PrP$^C$ substrate were supplemented with 25 μL of 10% PrP$^{0/0}$ brain homogenate (homogenized in PBS, 1% Triton X-100), 1 μL 500 mM EDTA pH 8.0, 10 μL of imidazole buffer (500 mM imidazole pH 7.0, 20mM MOPS pH 7.0, 150 mM NaCl, 0.75% Triton X-100), 4 μL PBS/1.25% Triton X-100, and 10 μL of PrP$^{Sc}$ seed. Reactions were sonicated with 20 sec pulses every 30 min.

## Preparation of M109 cofactor recPrP$^{Sc}$ and M109 protein-only recPrP$^{Sc}$ by sPMCA

M109 cofactor recPrP$^{Sc}$ and M109 protein-only recPrP$^{Sc}$ were generated by sPMCA based on a previously established protocol[16]. Expression and purification of bank vole PrP M109 recPrP 23–231 was performed as previously described[34]. sPMCA reactions were performed using a previously established protocol with minor modifications[34]. Two-hundred microliter reactions containing 6 μg/mL mouse recPrP 23–230 in conversion buffer (20 mM Tris pH 7.5, 135 mM NaCl, 5 mM EDTA pH 7.5, 0.15% (v/v) Triton X-100) were supplemented with purified brain-derived phospholipid cofactor[11] for cofactor recPrP$^{Sc}$ propagation or water for protein-only recPrP$^{Sc}$ propagation. All sPMCA reactions were sonicated with 15 sec pulses every 30 min.

## sPMCA with brain homogenate substrate

Brains, perfused with PBS containing 5mM EDTA, were harvested from either bank voles with the M109 genotype, WT C57BL/6J mice, PrP$^{0/0}$ mice, or WT Syrian hamsters. A 10% (w/v) brain homogenate was prepared initially by Potter homogenization in PBS. The crude homogenate was spun at 400 x *g* for 1 min, and the supernatant was removed and kept. Triton X-100 was added to the supernatant for a final concentration of 1% (v/v), and the supernatant was solubilized on ice for 10 min. Reactions contained 90 μL of brain homogenate and 10 μL of seed. Reactions were sonicated with 20 sec pulses every 30 min.

## sPMCA using immunopurified PrP$^C$

Reconstituted sPMCA experiments were adapted from Piro *et al.*[48]. Briefly, 150 μL reactions containing 20 μg/mL immunopurified bank vole M109 PrP$^C$ in conversion buffer [20 mM MOPS pH 7.0, 0.075% Triton X-100, 50 mM imidazole pH 7.0, 5 mM EDTA pH 7.5, 0.1 M

NaCl] were supplemented with either 10% (w/v) $PrP^{0/0}$ brain homogenate in PBS with 1% (v/v) Triton X-100, purified brain-derived phospholipid cofactor[11], PBS and 1% (v/v) Triton X-100, or 60 μg/mL polyadenylic acid potassium salt (Sigma Aldrich) and seeded with 15 μL $PrP^{Sc}$. Reactions were sonicated with 20 sec pulses every 30 min.

## RNase treatment and RNA quantification

Where indicated, crude brain homogenate substrates were pretreated using the following protocol adapted from[25]. In sPMCA experiments, all brain homogenates used for positive control reactions were mock-incubated under identical conditions in the absence of enzyme. Digestion with RNase, DNase-free, High Concentration (Sigma-Aldrich) was performed by incubation of 400 μL of brain homogenate with 5 μL of 10 mg/mL enzyme for 1 hr at 37°C. To purify RNA for quantification, 20 μL of brain homogenate was combined with 200 μL of TRIzol Reagent (Thermo Fisher Scientific) and purified using a Direct-zol RNA Miniprep kit (Zymo Research, Irvine, CA, USA). RNA was purified per the manufacturer's protocol, with the exception that 30 μL was used as the final elution volume instead of 50 μL. RNA concentration was quantified by spectroscopy using a NanoDrop 2000 (Thermo Fisher Scientific).

## Detection of $PrP^{Sc}$ in sPMCA reactions

Formation of $PrP^{Sc}$ was monitored by digestion of sPMCA samples with Proteinase K (PK) (Roche, Basel, Switzerland) and western blotting. Samples were digested with 64 μg/mL PK at 37°C with shaking at 750 r.p.m. Samples from sPMCA reactions using recPrP as the substrate were treated for 30 min, while samples using brain homogenate or immunopurified $PrP^{C}$ as the substrate were treated for 60 min. Digestion reactions were quenched by adding SDS-PAGE loading Buffer And heating to 95°C for 15 min. SDS-PAGE and western blotting were performed as described previously[48] using mAb 27/33. 20 μL of a sPMCA reaction was subjected to PK digestion. The minus PK (-PK) lane shown in each western blot Fig is used to determine the conversion efficiency of a sPMCA reaction (The amount of $PrP^{C}$ in the original substrate relative to the amount that was converted to $PrP^{Sc}$ during one round of PMCA). For crude brain homogenate sPMCA reactions, the -PK lane contains the same volume (20 μL) of a sPMCA reaction as a PK-digested sample. For sPMCA reactions using immunopurified $PrP^{C}$ or partially purified $PrP^{C}$, the–PK lane contains one-tenth the volume (2 μL) of a sPMCA reaction as a PK-digested sample due to the lower expected conversion efficiencies of these reactions.

## Ethics statement

Bank voles were obtained from the breeding colony at the Istituto Superiore di Sanità (ISS), Italy. The research protocol, approved by the Service for Biotechnology and Animal Welfare of the ISS and authorized by the Italian Ministry of Health, adhered to the guidelines contained in the Italian Legislative Decree 116/92, which transposed the European Directive 86/609/EEC on Laboratory Animal Protection The research protocol was performed under the supervision of the Service for Biotechnology and Animal Welfare of the ISS and was approved by the Italian Ministry of Health (decree number 1119/2015-PR).

## Supporting information

**S1 Fig. Lesion profiles of wild type bank voles infected with 139H and RML.** Lesion profiles of bank voles (M109 genotype) infected with 139H strain compared with RML strain, following third passage. The specific brain scoring areas analyzed were: (1) medulla, (2) cerebellum,

(3) superior colliculus, (4) hypothalamus, (5) thalamus, (6) hippocampus, (7) septum, (8) retrosplenial and adjacent motor cortex, and (9) cingu- lated and adjacent motor cortex.
(TIF)

**S2 Fig. Silver stain analysis of immunopurified bank vole PrP$^C$ substrate.** Twelve percent SDS/PAGE showing (from left to right): crude, detergent-solubilized bank vole brain homogenate (BV BH); immunopurified BV PrP$^C$ from bank vole brains; and molecular weight markers (ladders).
(TIF)

**S3 Fig. Purified PrP$^C$ substrates with specific glycoforms.** Western blot showing partially purified PrP$^C$ substrates from the indicated species that are used in sPMCA reactions. UN, PrP$^C$ substrate created by enzymatic deglycosylation of the DI substrate; DI, PrP$^C$ substrate eluted off the wheat-germ agglutinin column containing primarily diglycosylated PrP$^C$; ALL, PrP$^C$ substrate containing all three glycoforms.
(TIF)

**S4 Fig. Biological replicates of bank vole UN PrP$^C$ seeded with 139H.** Western blots showing additional three-round sPMCA reactions demonstrating the MW shift observed in Fig 6, row 4, righthand column. The red lines highlight a shift in the apparent MW of the day three sample. Day 0 samples are a seeded reaction not subject to sonication. -PK = samples not subjected to proteinase K digestion; all other samples were proteolyzed.
(TIF)

**S5 Fig. Effect of RNA on serial propagation of phospholipid cofactor-adapted PrP$^{Sc}$ conformer.** Three-round sPMCA reactions using mouse recombinant (rec)PrP substrate, mouse cofactor recPrP$^{Sc}$ seed, and purified phospholipid cofactor were performed as previously described[16], in the presence of varying concentrations of synthetic poly(A) RNA, as indicated. In the absence of RNA, cofactor PrP$^{Sc}$ maintains an ~18 kDa PK-resistant core during all 3 rounds of sPMCA. At [RNA] = 0.5 μg/mL, the PK-resistant core appears to shift stepwise to ~16 kDa between rounds 1–3; at [RNA] = 5 μg/mL, PrP$^{Sc}$ propagation appears to be completely inhibited; and at [RNA] = 50 μg/mL, the PK-resistant core appears to shift to ~16 kDa immediately during the first round of sPMCA. Thus, addition of RNA appears to either (1) inhibit propagation and/or (2) force conformational adaptation of cofator PrP$^{Sc}$ into a self-propagating conformer (similar to non-infectious protein-only PrP$^{Sc}$) in a concentration-dependent manner.
(TIF)

**S1 Table. Quantification of RNA in crude brain homogenate samples used for sPMCA.** Table showing RNA levels in RNA minipreps from untreated (-RNase) or RNase-treated (+-RNase) crude 10% brain homogenate substrates from various species, as measured by spectroscopy.
(DOCX)

## Acknowledgments

We would like to thank Tamutenda Chidawanyika and Therese Gerbich for their careful review of the manuscript and thoughtful advice and Ta Yuan Chang and Cathy Chang for their generous sharing of resources.

## Author Contributions

**Conceptualization:** Cassandra M. Burke, Surachai Supattapone.

**Formal analysis:** Cassandra M. Burke.

**Funding acquisition:** Cassandra M. Burke, Surachai Supattapone.

**Investigation:** Cassandra M. Burke, Daniel J. Walsh, Kenneth M. K. Mark, Nathan R. Deleault, Umberto Agrimi, Michele A. Di Bari.

**Methodology:** Cassandra M. Burke, Daniel J. Walsh, Nathan R. Deleault, Koren A. Nishina.

**Project administration:** Surachai Supattapone.

**Resources:** Daniel J. Walsh, Koren A. Nishina, Umberto Agrimi, Michele A. Di Bari, Surachai Supattapone.

**Supervision:** Surachai Supattapone.

**Writing – original draft:** Cassandra M. Burke, Surachai Supattapone.

**Writing – review & editing:** Cassandra M. Burke, Daniel J. Walsh, Kenneth M. K. Mark, Nathan R. Deleault, Surachai Supattapone.

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
