## [Decision Letter · Decision Letter 0]

18 Mar 2020

Dear Dr. Supattapone,

Thank you very much for submitting your manuscript "Strain Conformation Dictates Both Cofactor and Glycosylation Preferences for Prion Conversion" for consideration at PLOS Pathogens. As with all papers reviewed by the journal, your manuscript was reviewed by members of the editorial board and by several independent reviewers. The reviewers appreciated the attention to an important topic. Based on the reviews, we are likely to accept this manuscript for publication, providing that you modify the manuscript according to the review recommendations.

All three reviewers shared enthusiasm for the importance of the problem and your well organized and executed approach in addressing it. The reviews also had concerns regarding the interpretation of the work based on the data presented. Please take these concerns into consideration during your revisions. Thank you.

Sincerely,

Jason C. Bartz

Associate Editor

PLOS Pathogens

Neil A. Mabbott

Section Editor

PLOS Pathogens

Kasturi Haldar

Editor-in-Chief

PLOS Pathogens

orcid.org/0000-0001-5065-158X

Michael Malim

Editor-in-Chief

PLOS Pathogens

orcid.org/0000-0002-7699-2064

Dear Dr. Supattapone,

All three reviewers shared enthusiasm for the importance of the problem and your well organized and executed approach in addressing it. The reviews also had concerns regarding the interpretation of the work based on the data presented. Please take these concerns into consideration during your revisions. Thank you.

Take care,

Jason

Reviewer Comments (if any, and for reference):

Reviewer's Responses to Questions

**Part I - Summary**

Reviewer #1: The ms. by Burke et al. from a laboratory that has contributed much to the understanding of the process of PrP conversion to PrPSc, continues this line of investigation by seeking to determine the degrees to which prion strain propagation is controlled more by substrate factors other than primary structure (i.e. PrPc conformation and glycosylation) or by strain seed (PrPSc) conformation. An in vivo study in bank voles and mice inoculated with fixed mouse and hamster prion strains is used to establish that voles are susceptible to both mouse and hamster prion strains and that there are resultant strain-specific neuropathology patterns.

The issue of substrate environment vs. seed is approached in a complicated series of experiments using sPMCA, hamster and mouse strains as PrPSc seeds, and bank vole PrPc a universal substrate, compared in several experiments with hamster or mouse PrPc, including unglycosylated or infinity-purified versions thereof.

The above studies are well done and the manuscript carefully written. The series of experiments employing multiple strains and substrates, that are (for this reader at least) relatively arduous to track, keep in mind, and assemble in mind and lead to much re-reading to link the data to the concluding sentence(s). But clearly shown is that the fixed strains of mouse and hamster prions studied exhibit cofactor preferences. The chief concern (this reviewer) has is the use and interpretation of dictation and that strain seeds are dictators. It could be viewed, as nicely acknowledged in the Discussion, that the (mouse or hamster-adapted) strains (of scrapie) have arisen from environments that have led to the strain seed conformation selection and fixation (usually involving serial passage). In that respect, have the substrate environments, usually brain tissue, not had part in the dictating that gave rise to the strains? So to this reviewer casting prion strain/seed, vs. substrate properties, as the dictator seems overly strained.

What is elegantly shown in that the interaction of seed, substrate conformations, glycosylation, and cofactors (all found cellular environments) determine the evolution of variants and eventual dominance of prion strains (the classic example used, 139A and H). [Perhaps the most robust naturally occurring example in BSE, which can ‘dictate’ across species boundaries.] So (to this reviewer) the process of strain origin and dominance seems analogous to fitness selection without genetic mutation to cement the process--certainly remarkable and why the protein only hypothesis is so significant (the point the ms. addresses). Therefore (again, to this reviewer), the work is excellent, but the emphasis on seed dictation is overstated.

Reviewer #2: Burke and colleagues provide a nice study of strain-dependent effects of different cofactors and glycosylation status of PrPC in PMCA-based PrP conversion. The work addresses the fundamental issue of what factors control the faithful propagation, and adaptation, of prion strains. For the most part the experiments are well done, convincing, and clearly explained. Overall the study makes a valuable contribution to the field. However, I have a few concerns and suggestions to improve the manuscript.

Reviewer #3: The study by Burke et al. addresses an important question in the prion field, that is the molecular basis underlying species-specific preferences for prion conversion. The authors conclude that species-specific cofactor and glycosylation preferences are determined by the conformation of the PrPSc seed, rather than by the amino acid sequence of the PrPC substrate. In general terms, the work is not 100% new, as other works indicated the involvement of cofactor molecules and glycosylation of PrPC in maintaining the infectivity and strain properties in prions. However, this study is well organized and well performed, and therefore it qualifies for publication in Plos Pathogens, after the authors have addressed the points indicated below.

Major concerns:

1. The authors state that distinct prion strains (RML and 139H) faithfully propagate in the presence of cofactors that are specific for each strain. However, prion strains are operationally defined by differences in heritable phenotypes under controlled experimental setups, like tissue tropism, incubation time and lesions profile. The authors should inoculate the PMCA products amplified in the presence of the specific cofactors into bank voles, to show that they retain the same lesion profiles of the original seeds.

2. The authors propose that the phenomenon of neurotropism could be explained by the preference of PrPSc seed for specific cofactors that are enriched only in some brain regions. I think that, although it might be a reasonable hypothesis, the data are not strong enough to support this claim. If feasible, the authors could replicate some of their experiments (i.e. PMCA amplification of seeds with and without RNAse treatment) using only some specific regions of the bankvole brains, instead of the whole brain homogenate, for example the brain region where these specific strains are known to preferentially propagate.

3. The authors mention a very interesting point (Discussion, Lines 234-243) about how cofactors determine the preference in conversion of prion seeds, and propose that molecules serving as cofactors for one prion strain may actually inhibit conversion of another strain. It would further strengthen the mechanism of this study if validation on this hypothesis could be performed via relative experiments, while it’s not contained in the required points.

**Part II – Major Issues: Key Experiments Required for Acceptance**

Reviewer #1: The concerns deal more with interpretation and statement thereof than the experiments per se'. As described in the reviewer comments above.

Reviewer #2: 1. The PMCA amplifications of the mouse prion strains in Fig 2 are not convincing in that the banding profiles are a rather faint and blurry and do not differ much at all from the unsonicated “0 round” samples whether the substrate BH for the reaction was RNase treated or not. The data in subsequent figures, using purified PrPC, are much more convincing. It would help the reader if the authors provided more perspective on why we should believe what is going on in Fig 2.

2. The authors’ main thesis is that prion conformation, rather than sequence, is what controls cofactor and glycoform selection. However, a striking feature in the data overall is that all of the murine prion strains that were tested behave one way in terms of cofactor preference (preferring PE) while the hamster prion strains behave another (needing RNA). This would suggest that the PrP sequence of the prion (i.e. mouse vs hamster), in addition to different conformers thereof, are highly influential. If, as the authors argue, sequence doesn’t matter (much), why can none of the mouse strains (with their different conformations) use RNA as a cofactor for propagation, while both of the hamster strains can?

Reviewer #3: Technical concerns:

1. In Fig.3, the 1st passage of RML seed for Mo PrPC substrate shows a weak propagation of PrPSc in the “+ RNA group”, which could also be found in “-cofactor” groups for 1st passages of 139H seed for Ha PrPC, RML seed for BV PrPC, and 139H seed for BV PrPC. Please discuss the possible reasons.

**Part III – Minor Issues: Editorial and Data Presentation Modifications**

Reviewer #1: The Results section contains extensive use of acronyms. While defined initially, the density of these can make sentences hard to interpret without back-searching to be sure of the what is being studied/tested. More information to help the reader with recalling the identity of the strains when mentioned would help the reader put the experiment premise and result in context without backtrack reading to remember what e.g. 139A, Me7, 22A and 301C are again. Just adding mouse helps a lot.

Reviewer #2: L210: check spelling

L293: Suggest Lawson & Priola, EMBO J (2001) as a primary reference here.

Reviewer #3: (No Response)

PLOS authors have the option to publish the peer review history of their article (what does this mean?). If published, this will include your full peer review and any attached files.

Reviewer #1: No

Reviewer #2: Yes: Byron Caughey

Reviewer #3: No
---

## [Editor Report · Decision Letter 1]

24 Mar 2020

Dear Dr. Supattapone,

We are pleased to inform you that your manuscript 'Cofactor and Glycosylation Preferences for In Vitro Prion Conversion Are Predominantly Determined by Strain Conformation' has been provisionally accepted for publication in PLOS Pathogens.

Best regards,

Jason C Bartz

Associate Editor

PLOS Pathogens

Neil Mabbott

Section Editor

PLOS Pathogens

Kasturi Haldar

Editor-in-Chief

PLOS Pathogens

orcid.org/0000-0001-5065-158X

Michael Malim

Editor-in-Chief

PLOS Pathogens

orcid.org/0000-0002-7699-2064
---

## [Editor Report · Acceptance letter]

6 Apr 2020

Dear Dr. Supattapone,

We are delighted to inform you that your manuscript, "Cofactor and Glycosylation Preferences for In Vitro Prion Conversion Are Predominantly Determined by Strain Conformation," has been formally accepted for publication in PLOS Pathogens.

Best regards,

Kasturi Haldar

Editor-in-Chief

PLOS Pathogens

orcid.org/0000-0001-5065-158X

Michael Malim

Editor-in-Chief

PLOS Pathogens

orcid.org/0000-0002-7699-2064